# ToMAP: Training Opponent-Aware LLM Persuaders with Theory of Mind

Peixuan Han [1]   Zijia Liu [2][1]   Jiaxuan You [1]

## Abstract

Large language models (LLMs) have shown promising potential in persuasion, but existing works on training LLM persuaders are still preliminary. Notably, while humans are skilled in modeling their opponent's thoughts and opinions proactively and dynamically, current LLMs struggle with such Theory of Mind (ToM) reasoning, resulting in limited diversity and opponent awareness. To address this limitation, we introduce Theory of Mind Augmented Persuader (**ToMAP**), a novel approach for building more flexible persuader agents by incorporating two theory of mind modules that enhance the persuader's awareness and analysis of the opponent's mental state. Specifically, we instruct the persuader to consider possible objections to the target claim, and train a module to predict the opponent's agreement level on these objections. Experiments show that the ToMAP persuader, while containing only 3B parameters, outperforms much larger baselines, like GPT-4o, with a relative gain of 39.4% across multiple persuadee models and diverse corpora. Notably, ToMAP exhibits complex reasoning chains and reduced repetition during training, which leads to more diverse and effective arguments. These results underscore ToMAP's potential for developing more persuasive language agents. Code is available at: https://github.com/ulab-uiuc/ToMAP.

## 1. Introduction

*"If you know the enemy and know yourself, you need not fear the result of a hundred battles."*

— *Sun Tzu, The Art of War*

Persuasion—the process of influencing others' attitudes,

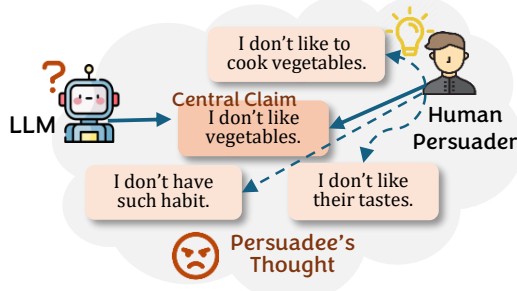

Figure 1. **Human thoughts about claims are inherently interconnected.** While human persuaders can recognize this network of related propositions, LLMs often focus narrowly on the central claim alone and omit other claims.

beliefs, or behaviors—is fundamental to human communication and social interaction, and is recognized as a complex and advanced social behavior. Recently, large language models (LLMs) have also begun to engage in this cognitively demanding practice. State-of-the-art LLMs like GPT-4o have demonstrated remarkable persuasive abilities, which can generate persuasive arguments (Bai et al., 2023; Palmer & Spirling, 2023) and influence human attitudes in specific domains (Karinshak et al., 2023; Takayanagi et al., 2025). At the same time, as LLMs become increasingly integrated into diverse social and professional domains like customer servicing (Amazon Web Services, 2023) and academic reviewing (AAAI, 2025), effectively persuading an LLM has also grown in importance.

Despite the advances in LLM persuasion, current LLM-based persuaders face significant limitations in modeling the interactive and dynamic nature of persuasion. During a conversation, human participants naturally consider related claims around the central claim, forming a rich cognitive graph as shown in Figure 1. Intuitively, skilled human persuaders navigate this cognitive graph, analyzing the persuadee's mental state and identifying "nodes" in the cognitive graph that are ripe for elaboration and could potentially affect the central claim. This process relies on the ability to model and reason about others' beliefs, a cognitive capacity known as **theory of mind** (ToM). As a cornerstone of human social cognition, ToM enables humans to tailor their messages to the opponent's perspectives dynamically, making persuasion more effective and tailored to their opponents.

[1]University of Illinois Urbana-Champaign [2]Tongji University. Correspondence to: Peixuan Han <ph16@illinois.edu>, Jiaxuan You <jiaxuan@illinois.edu>.

*Proceedings of the 43rd International Conference on Machine Learning*, Seoul, South Korea. PMLR 306, 2026. Copyright 2026 by the author(s).

LLMs, by contrast, often overlook the interconnected structure of the cognitive graph and rarely consider the opponent's perspective in dynamic exchanges. This deficiency in theory of mind leads to two key shortcomings in current LLM-based persuaders. First, after stating their position in the initial round, they tend to repeat the same arguments rather than drawing on new insights from the broader cognitive graph (Chu et al., 2024). For instance, the persuader trained without ToM modules exhibits up to 12% 8-gram content overlap between consecutive turns (Figure 3b). Second, LLM-based persuaders are often rigid and self-centered, focusing narrowly on their own arguments rather than adapting to the opponent's views dynamically (Wang et al., 2025b; Becker, 2024). This lack of flexibility is exemplified in Figure 5, where baseline models employ opponent-oriented strategies far less frequently.

To address the above limitations, we propose **ToMAP** (**T**heory **o**f **M**ind **A**ugmented **P**ersuader), a novel training framework inspired from human theory of mind process that consists of two dedicated modules. Firstly, we introduce the **counterclaim predictor**, which prompts the persuader to model relevant claims in the persuadee's cognitive graph explicitly and proactively anticipate potential objections. By explicitly predicting counterclaims, this module enables the persuader to plan ahead, generate more diverse arguments, and address objections preemptively. Furthermore, we design the **opponent attitude predictor**, which estimates the persuadee's current level of agreement on the counterclaims in the cognitive graph. By evaluating the persuadee's mental state and identifying which of the opponent's claims are firmly held or uncertain, the attitude predictor plays a key role in enabling audience-aware and adaptable persuasion strategies. Finally, we employ **reinforcement learning** (RL) to help the persuader interpret and leverage these insights effectively, since the base model cannot effectively utilize the ToM information directly.

Our experiments demonstrate that **the ToMAP model**, with only 3B parameters, **significantly outperforms baseline configurations and exhibits persuasiveness comparable to or exceeding that of several state-of-the-art LLMs** across diverse corpora and against multiple persuadee models. Further analyses reveal that ToMAP produces more complex reasoning chains and reduces repetition during training, achieves steady persuasion gains in long conversations, and adopts more logical and opponent-aware strategies. These findings highlight ToMAP's unique strengths robust and effective framework for developing more persuasive LLM agents, suggesting ToM as a promising directions for building more persuasive language agents.

## 2. Related Work

**LLM Persuasion.** LLM persuasion has attracted increasing attention with the rapid development of state-of-the-art LLMs (Bozdag et al., 2025b; Jaech et al., 2024; Rogiers et al., 2024; Breum et al., 2024), showing great potentials on areas like political issues (Bai et al., 2023; Palmer & Spirling, 2023; Potter et al., 2024), vaccine promotion (Karinshak et al., 2023), investment decisions (Takayanagi et al., 2025), and factual knowledge (Costello et al., 2024). Empirical evidence also shows that LLMs can potentially shift human attitudes in real-world conversations (Salvi et al., 2024; Shi et al., 2020b; Potter et al., 2024). Several systematic frameworks have been proposed to systematically evaluate the persuasiveness of LLMs (Pauli et al., 2024; Salvi et al., 2024; Bozdag et al., 2025a; Singh et al., 2024; Durmus et al., 2024). (Carrasco-Farre, 2024) further found the persuasiveness of LLMs comes from proposing arguments that require higher cognitive effort.

To enhance the persuasion capabilities of LLMs, researchers utilize humans' strategies(Wang et al., 2019; Yang et al., 2019) and high-quality dialogues(Singh et al., 2024; Stengel-Eskin et al., 2024; Jin et al., 2024; Furumai et al., 2024) to train LLMs. Recently, advanced agentic pipelines building upon fixed LLMs have also been developed for the persuasion task, such as TreeDebater (Wang et al., 2025a), Debate-to-Write (Hu et al., 2024), and AI-Realtor (Wu et al., 2025a). Reinforcement learning (RL) for persuasion has primarily focused on narrow tasks, such as negotiation games (Keizer et al., 2017; Lewis et al., 2017) or consistency checking (Shi et al., 2020a). Concurrent to our work, (Cheng & You, 2025) explores training general persuasion models with RL.

**Theory of Mind.** Theory of Mind (ToM) refers to the ability to understand others by attributing mental states (Wimmer & Perner, 1983; Baron-Cohen et al., 1985; Wellman, 2018). ToM ability in AI systems has attracted great attention (Cuzzolin et al., 2020; Langley et al., 2022; Williams et al., 2022; Schossau & Hintze, 2023), and researchers have proposed benchmarks to evaluate such capabilities (Kim et al., 2023; Le et al., 2019; Wu et al., 2023). More complex ToM tasks are also introduced, showing future directions of enhancing LLMs' ToM abilities (Xu et al., 2024; He et al., 2023; Street et al., 2024).

Beyond evaluating LLMs' ToM capabilities, researchers have also explored ways to enhance such abilities, which can be broadly categorized into two lines: explicit and implicit methods. Explicit methods involve asking questions to reveal the target agent's intentions (Wu et al., 2025b; Kobalczyk et al., 2025; Qian et al., 2024), typically to facilitate human–computer interaction or problem solving. Implicit methods, by contrast, aim to infer the target agent's mental state from the conversation alone and are commonly used in scenarios where communication is limited or costly. Approaches that facilitate implicit ToM abilities include Bayesian inverse planning (Kim et al., 2025; Zhang et al., 2025), test-time simulation (Wilf et al., 2024), and agentic

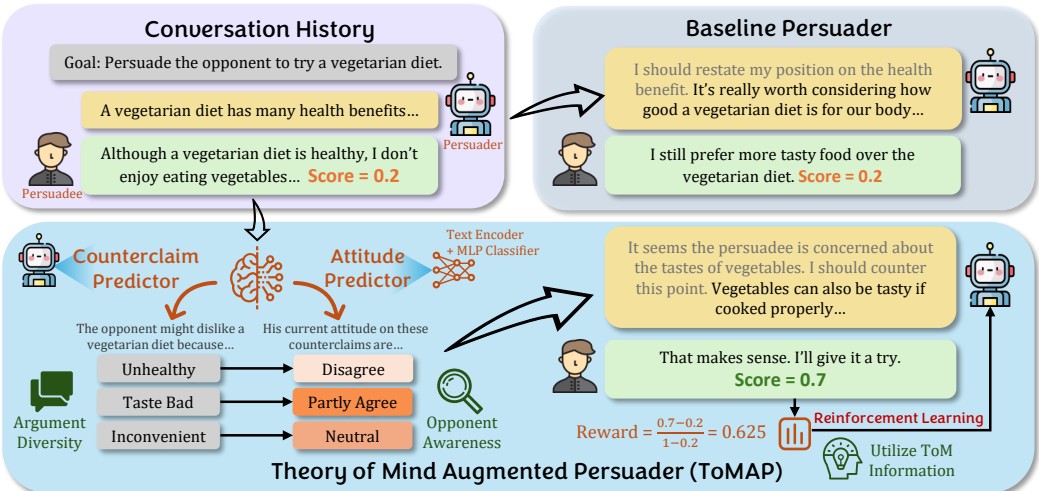

*Figure 2.* **Overview of Theory of Mind Augmented Persuader (ToMAP)**. ToMAP utilizes two ToM modules, the counterclaim predictor and attitude predictor, to effectively model the opponent's mental state during the conversation. This design enables ToMAP to provide more diverse arguments and counter the persuadee's concerns in a more flexible and opponent-aware manner.

finetuning (Jung et al., 2024; Hwang et al., 2025), which is adopted in this work. Persuasion, in particular, is a task that requires implicit ToM understanding—since the two agents are adversarial rather than cooperative—and its close relationship with Theory of Mind has been widely studied (Peterson et al., 2018; McAlister & Cornwell, 2009; Slaughter et al., 2013). More recently, (Yu et al., 2025) evaluated LLMs' ToM in persuasive dialog, and (Zhang & Zhou, 2025) identified the lack of ToM in LLM conversations, highlighting the importance of ToM in persuasion.

## 3. Methodology

In this section, we introduce the key components of Theory of Mind Augmented Persuader (ToMAP). We first describe the setting of the persuasion task in Section 3.1 and the reinforcement learning algorithm used to train the persuader in Section 3.2. Finally, we introduce two theory of mind modules that provide critical information about the opponent's thoughts Section 3.3.

### 3.1. The Setting of Persuasion Task

The persuasion process is a multi-turn conversation between two LLM agents: a persuader, denoted as $\mathbf{X}$, and a persuadee, denoted as $\mathbf{Y}$. During the conversation, the persuader aims to convince the persuadee to support a given claim $Q$. Formally, we denote the $i$th turn in the conversation as $T_i$, and the conversation history till the $i$th turn as $H_i = T_{1\ldots i}$[1]. Since the two agents take turns speaking in

---

[1]Specially, we define $H_0$ as an empty string.

the conversation, the next turn can be represented as:

$$T_n \sim \begin{cases} P_{\mathbf{X}}(\cdot \mid p_{\text{debate}}, Q, H_{n-1}) & \text{if } n \text{ is odd,} \\ P_{\mathbf{Y}}(\cdot \mid p_{\text{debate}}, Q, H_{n-1}) & \text{if } n \text{ is even.} \end{cases} \quad (1)$$

After each persuadee's turn, we employ a 5-point Likert scale (Likert, 1932) to assess the persuadee's perspectives through prompting, which can be represented as:

$$s'(\mathbf{Y}, H_n, Q) = P_{\mathbf{Y}}(\cdot \mid p_{\text{attitude}}, Q, H_n) \in \{0, 1, 2, 3, 4\}. \quad (2)$$

In above formulas, $p_{\text{debate}}$ and $p_{\text{attitude}}$ are task-specific prompts, specified in Section A.

Since LLMs are known to exhibit inconsistency in opinions, which means a LLM might agree to both one claim and the opposite claim (Jung et al., 2022; Liang et al., 2024), we consider the persuadee's opinion on two contradictory claims, and calculate the balanced agreement score as[2]:

$$S(\mathbf{Y}, H_n, Q) = 0.5 + \frac{s'(\mathbf{Y}, H_n, Q) - s'(\mathbf{Y}, H_n, \neg Q)}{8} \in [0, 1], \quad (3)$$

where $\neg$ **means the opposition of the original claim**. According to the formula, the $S$ score increases when either 1) the LLM expresses stronger agreement with $Q$; or 2) the LLM expresses stronger disagreement with $\neg Q$. We believe the persuadee's self-reported attitude serves as a scalable and reliable evaluation metric for two main reasons. First, as LLM agents are being deployed into practical applications frequently and "persuading an LLM" is a practical task in domains like customer service and paper review. Second, as shown in Section 5.1, LLMs and humans demonstrate a high degree of consistency in their judgments of persuasion.

---

[2]We have $S(\mathbf{Y}, H_n, Q) + S(\mathbf{Y}, H_n, \neg Q) = 1$, which ensures the consistency of LLM opinion judgment.

## 3.2. A Reinforcement Learning Formulation for Persuader Training

Reinforcement learning (RL) has demonstrated great success in recent work such as Deepseek-R1 (Guo et al., 2025), Search-r1 (Jin et al., 2025) and RM-R1 (Chen et al., 2025), where models develop complex reasoning skills. To incentivize LLMs' persuasion ability without extensive labeled data, we also utilize RL to train the persuader through conversations and feedback from the persuadee.

As we find directly optimizing the persuader in multi-round interaction is challenging and requires much more complex design (Jin et al., 2025; Wang et al., 2025c), we propose decomposing the multi-turn conversations as single-turn activities to enhance the stability and efficiency of our training schema. Specifically, given the conversation history $H$ and target claim $Q$, we optimize the persuader to generate the next argument that maximizes the attitude shift of the persuadee. Formally, at the persuader's turn $2k - 1$, we use

$$\text{diff}(T_{2k-1}, Q) = S(\mathbf{Y}, H_{2k}, Q) - S(\mathbf{Y}, H_{2k-2}, Q) \quad (4)$$

to measure the attitude shift after the persuader's turn and the persuadee's following turn. Then, we normalize the agreement gap to the range of $[-1, 1]$ to obtain the persuasion reward:

$$r_{\text{persuade}}(T_{2k-1}, Q) = \begin{cases} \dfrac{\text{diff}(T_{2k-1}, Q)}{1 - S(\mathbf{Y}, H_{2k-2}, Q)} & \text{if } \text{diff}(T_{2k-1}, Q) \geq 0^3, \\[2ex] \dfrac{\text{diff}(T_{2k-1}, Q)}{S(\mathbf{Y}, H_{2k-2}, Q)} & \text{if } \text{diff}(T_{2k-1}, Q) < 0. \end{cases} \quad (5)$$

Notably, our reward design captures the intuition that boosting agreement on claims already having a high score is more challenging. It promotes persuasive strategies that effectively reinforce claims that are already partially accepted. In addition to the persuasion reward, we also employ a set of auxiliary rewards to finely regulate formatting and control generation quality, as detailed in Section C.

We adopt proximal policy optimization (PPO) (Schulman et al., 2017), a widely-used and stable algorithm, to train the persuader policy $\mathbf{X}$, optimizing it with the standard PPO objective function:

$$\mathcal{J}_{\text{PPO}}(\mathbf{X}) = \mathbb{E}_{x \sim \mathcal{D}, T \sim \mathbf{X}} \Big[ r(T, Q) - \beta \mathbb{D}_{\text{KL}} \big( \mathbf{X}(T \mid x) \,\big\|\, \mathbf{X}_{\text{ref}}(T \mid x) \big) \Big] \quad (6)$$

where $D$ is the dataset, $r$ is the reward function, $\beta$ is a hyperparameter controlling KL penalty and $\mathbf{X}_{\text{ref}}$ is the reference policy of the persuader. Notably, $x = (p_{\text{debate}}, Q, H)$ contains the system prompt, target claim, as well as the conversation history.

---

[3] $S(\mathbf{Y}, H_{2k-2}) = S(\mathbf{Y}, H_{2k}) = 0$ is a special case where the formula is undefined. In this case, we define $r_{\text{persuade}}$ to be 0 because the turn has no impact on the persuadee's agreement.

## 3.3. Theory of Mind (ToM) Modules

Theory of Mind (ToM)—the ability to understand others' mental states, especially opinions and attitudes—is essential for effective persuasion. To equip LLMs with such ability and provide crucial information for their strategic persuasion, we incorporate **Counterclaim Predictor** and **Opponent Attitude predictor.** into our persuader agent, as illustrated in Figure 2.

### 3.3.1. COUNTERCLAIM PREDICTOR

Large language models (LLMs) often struggle to conceptualize a debate topic's associated claims, which reduces their ability to anticipate an opponent's beliefs and potential counterarguments. Therefore, the counterclaim predictor is proposed to encourage the persuader to proactively anticipate counterarguments. Based on that, the persuader can craft more contextually aware persuasive strategies.

**Implementation.** We use a delicately designed prompt to encourage the persuader to consider claims challenging the persuasion goal. Formally, the counterclaim predictor can be expressed as

$$\neg q_{1 \ldots k} \sim \mathbf{X}(p_{\text{counterclaim\_pred}}, Q, k), \quad (7)$$

where $Q$ is the original claim and $\neg q_{1 \ldots k}$ are the $k$ counterclaims against $Q$.

### 3.3.2. OPPONENT ATTITUDE PREDICTOR

Building upon the counterclaim predictor, we further propose the opponent attitude predictor to provide the persuader with a dynamic understanding of the persuadee's evolving beliefs, which are necessary for flexible and targeted persuasion in the subsequent turns.

**Implementation.** As zero-shot prompting doesn't yield satisfactory results, we use an external classifier, which consists of a text encoder and a multilayer perceptron (MLP), to predict opponent attitudes. Specifically, we use the pretrained BGE-M3 encoder (Chen et al., 2024) to encode the claim and the conversation as vectors separately. The two embedding are then concatenated and passed to a 5-way MLP classifier, which predicts the agreement score $s'$:

$$s'_{\text{pre}}(H, Q) = \mathbf{M}(\mathbf{E}(H) \| \mathbf{E}(Q)) \quad (8)$$

where $\mathbf{E}$ denotes the encoder, $\mathbf{M}$ denotes a multi-layer perceptron, $H$ is the conversation history and $Q$ is the claim. "$\|$" refers to the concatenation of vectors. With the cross-entropy loss, we train $\mathbf{M}$ to accurately predict the persuadee's attitudes on counterclaims generated by the counterclaim predictor. Refer to Section D.2 for details regarding the attitude predictor.

### 3.3.3. INCORPORATING TOM INFORMATION

Information from the ToM modules is included in the persuader's prompt during training and inference, serving as an

*Table 1.* **ToMAP is highly effective in persuasion**. Numbers represent the agreement shift in the conversation where two agents each take 3 turns. Qwen2.5-7B-Instruct is the persuadee used during training; the other two persuadees are out-of-domain tests. The best results among each base model are bolded.

| Persuader | Size | Persuadee: Qwen2.5 | | | Persuadee: LLaMa3.1 (**OOD**) | | | Persuadee: Phi-4 (**OOD**) | | | Avg. |
|---|---|---|---|---|---|---|---|---|---|---|---|
| | | CMV | Anthropic | args.me | CMV | Anthropic | args.me | CMV | Anthropic | args.me | |
| Gemma-3 | 27B | 13.52 | 10.02 | 12.21 | 11.35 | 7.54 | 9.69 | 33.09 | 29.91 | 28.07 | 17.27 |
| LLaMa3.1 | 70B | 12.58 | 5.27 | 7.09 | 2.23 | 4.10 | 2.22 | 19.84 | 20.50 | 21.29 | 10.57 |
| DS-R1 | 671B | 16.04 | 7.81 | 10.34 | 8.51 | 8.79 | 7.69 | 32.31 | 30.61 | 31.13 | 17.02 |
| GPT-4o | N/A | 13.36 | 7.62 | 9.51 | 3.35 | 3.90 | 2.50 | 23.97 | 26.56 | 22.06 | 12.54 |
| Qwen2.5 | 3B | 12.75 | 5.66 | 6.90 | 4.97 | 7.09 | 4.09 | 19.44 | **23.24** | 21.74 | 11.76 |
| +SFT | 3B | 13.41 | 7.95 | 8.97 | 4.34 | 7.96 | 6.51 | 17.95 | 22.35 | **22.98** | 12.49 |
| +RL | 3B | 17.42 | 12.92 | 13.77 | **12.15** | **13.59** | **12.89** | 16.28 | 16.84 | 15.51 | 14.60 |
| +ToMAP | 3B | **23.01** | **15.82** | **18.75** | 10.77 | 12.10 | 10.89 | **20.93** | 22.79 | 22.25 | **17.48** |
| LLaMa3.2 | 3B | 14.40 | 7.42 | 8.40 | 7.56 | 8.78 | 7.22 | 18.86 | 18.75 | 19.97 | 12.37 |
| +SFT | 3B | 14.90 | 7.44 | 9.10 | 6.18 | 10.20 | 8.89 | 16.41 | 20.03 | 20.95 | 12.68 |
| +RL | 3B | 21.34 | 16.40 | 19.34 | **19.16** | 16.21 | 17.62 | 25.69 | 30.27 | **29.87** | 21.76 |
| +ToMAP | 3B | **26.89** | **23.63** | **20.96** | 17.57 | **18.16** | **20.84** | **31.64** | **32.42** | 27.00 | **24.35** |

augmentation of input $x$ in Equation (6):

$$
\begin{aligned}
x_{\text{aug}} &= (x, \neg q_{1\ldots k}, s'_{\text{pre}}(H, \neg q_{1\ldots k})) \\
&= \left(p_{\text{debate}}, Q, H, \neg q_{1\ldots k}, s'_{\text{pre}}(H, \neg q_{1\ldots k})\right).
\end{aligned}
\tag{9}
$$

In conclusion, the ToMAP framework trains the persuader with ToM-aware reinforcement learning, enabling it to effectively incorporate the opponent's mental state information into its persuasive planning, resulting in more diverse and impactful arguments. We show the whole pipeline for ToM-enhanced persuasion in Section C.3.

## 4. Experiments

### 4.1. Experimental Settings

**Datasets.** During training, we use the Cornell CMV dataset (Tan et al., 2016), a widely used corpus[4] encompassing a broad range of topics, including political views, philosophy, and environment. In evaluation, we also included two other benchmarks: Anthropic Persuasion Dataset (Durmus et al., 2024) and the args.me corpus (Ajjour et al., 2019), to assess persuaders in out-of-domain persuasion topics. Notably, all topics collected are controversial issues, so we set the persuadee's initial stance to be negative in the prompt (i.e. $S(\mathbf{Y}, H_0, Q) < 0.5$; refer to Section E). Please refer to Section A for prompts, Section B for data statistics.

**Base Models.** We utilize two base models to train persuaders: Qwen2.5-3B-Instruct (Yang et al., 2024) and LLaMa3.2-3B-Instruct (Grattafiori et al., 2024). The persuadee model during training is Qwen2.5-7B-Instruct. During evaluation, we also evaluate persuaders against two

---

[4]The CMV dataset is derived from Reddit's ChangeMyView forum: https://www.reddit.com/r/changemyview.

different persuadee models to assess the generalization: LLaMa3.1-8B-Instruct and Phi-4 (Abdin et al., 2024). In Table 11, we show additional results on two larger persuadee models: Qwen3-Next-80B-A3B and GPT-4o-mini.

**Metric.** We use **agreement shift**, the gap between the persuadee's agreement score before and after the conversation, as the metric indicating persuasiveness. Formally, the agreement shift is calculated as $S(\mathbf{Y}, H_{2n}, Q) - S(\mathbf{Y}, H_0, Q)$ ($n = 3$ in the main experiment), which is as a percentage. During evaluation, we set the temperature to $1.0$ to encourage diverse generation, and report the average score over 3 trials to ensure evaluation stability.

**Experimental Settings.** There are four types of models that we evaluate:

• Off-the-shelf LLMs: Besides Qwen2.5-3B-Instruct and LLaMa3.2-3B-Instruct, we also introduce 4 larger models: Gemma-3-27B (Team et al., 2025), LLaMa3.1-70B-Instruct (Grattafiori et al., 2024), GPT-4o (Hurst et al., 2024) and Deepseek-R1 (Guo et al., 2025).

• SFT: we conduct inference on the training set, and select conversations where the base model successfully persuades the opponent (i.e., agreement shift $> 0$). These conversations are then used to finetune the base model.

• RL: we train the base model trained using RL without incorporating Theory of Mind (ToM) information. Please refer to Section C for details of the training procedure.

• ToMAP: the base model trained with RL and augmented by two ToM-based components: **Counterclaim Predictor** and **Opponent attitude predictor**. By default, 3 counterclaims from the opponent's side are generated and predicted attitudes on these claims are provided to the persuader.

*Table 2.* Comparison of average persuasion score (against Qwen2.5-7B on 3 datasets) with and without ToM modules. Encouraging long thought and adding ToM info have limited impact on base models' scores, showing that **RL is crucial for persuaders to utilize ToM information.**

| Persuader | Qwen2.5-3B | Gemma-3-27B | LLaMa3.1-70B | Deepseek-R1 | GPT-4o | Qwen2.5-3B+ToMAP |
|---|---|---|---|---|---|---|
| w/o ToM info | 8.44 | 11.92 | 8.31 | 11.40 | 10.17 | 14.47 |
| w/ Long CoT Prompt | 8.73 | 12.58 | 9.19 | 11.46 | 12.10 | N/A |
| w/ ToM info | 8.18 | 13.26 | 9.56 | 11.32 | 11.69 | **19.19** |

## 4.2. Main Results

Results in Table 1 highlight the following findings:

**RL is effective in enhancing persuasiveness.** The base model's influence on the opponent's opinion is very limited. Through training on carefully designed reward signals that encourage highly persuasive arguments, RL shows a remarkable relative performance gain of 17.78%. On the other hand, SFT doesn't yield satisfactory results as it's a static approach.

**ToM modules make stronger persuaders.** Equipped with ToM modules that analyse the opponent's thoughts, ToMAP shows a further relative gain of 26.14% compared to RL. The ToM modules allow for a deeper understanding of the context and the opponent, ultimately enhancing their persuasiveness. Notably, while ToMAP only contains 3B parameters, it outperforms much larger baselines[5], showing that small models can also be persuasive through proper training.

**ToMAP generalizes well in most scenarios.** Similar to humans, different LLM persuadees have varied preferences in persuasion, as exemplified in Section E. However, ToMAP achieves a rather stable performance. Among 18 settings, ToMAP outperforms RL on 13 and outperforms SFT on 17, indicating that the attitude predictor enables the persuader to learn more flexible persuasion strategies that persuade opponents dynamically.

Additionally, Section D show that the ToM modules themselves excel at modeling opponent thoughts, further validating our approach, and Section 5.5 shows ToMAP works well for a broader range of persuadees.

## 4.3. Ablation Studies

### 4.3.1. THE CRUCIAL ROLE OF RL

As described in Section 3.3, the counterclaim and attitude predictions are added to the persuader's prompt without requiring finetuning. A natural question is whether these ToM modules can improve the base model's zero-shot per-

*Table 3.* **Attitude prediction significantly enhances persuasiveness.** The persuadee here is Qwen2.5-7B.

| Setting | CMV | Anthropic | args.me |
|---|---|---|---|
| Qwen2.5-3B | 12.75 | 5.66 | 6.90 |
| +Long CoT | 14.25 | 6.25 | 7.66 |
| +ToMAP(w/o att) | 15.57 | 9.55 | 10.45 |
| +ToMAP-gt | **23.57** | **17.05** | 17.65 |
| +ToMAP | 23.01 | 15.82 | **18.75** |

formance. Our experiments show they cannot, as the persuadee's mental state information is only fully utilized after RL training.

From Table 2, we can clearly observe that the optimal performance is achieved only in the "ToMAP model with ToM information" setting. Notably, merely adding ToM information to the Base model without proper training brings no benefit. Larger models may utilize ToM information better in a zero-shot manner, but their gains with ToM are still limited, as they're not trained to utilize such information. In addition, adding prompts to encourage longer, more careful thoughts also has little impact on persuasion. Therefore, **both ToM modules and RL training are necessary for the performance gain**: ToM modules provide valuable assessments of the opponent, while RL enables the persuader to exploit this information and develop a better strategy.

### 4.3.2. THE CRUCIAL ROLE OF THE ATTITUDE PREDICTOR

In this section, we analyse the role of the attitude predictor. Specifically, we compare ToMAP with two variants that are trained with different attitude predictors: "ToMAP(w/o att)" only uses the counterclaim predictor during training[7], and "ToMAP-gt" uses the ground truth attitudes obtained by actually querying the persuadee after each turn, which is a idealized scenario where the persuader completely understands the persuadee's thoughts.

From Table 3, we can find that ablating the attitude predictor largely hurts performance, demonstrating the critical role of effectively modeling opponent attitude. In addition, the gap between ToMAP and ToMAP-gt, the variant with perfect theory of mind information (the upper bound), is marginal,

---

[5]ToMAP's advantages over Gemma-3 and Deepseek-R1 are marginal; however, given the significantly larger parameter counts of these models, the results still highlight ToMAP's impressive persuasiveness.

[7]Because the attitude predictor is built on top of the counterclaim predictor, we can ablate the attitude predictor alone but not the counterclaim predictor alone.

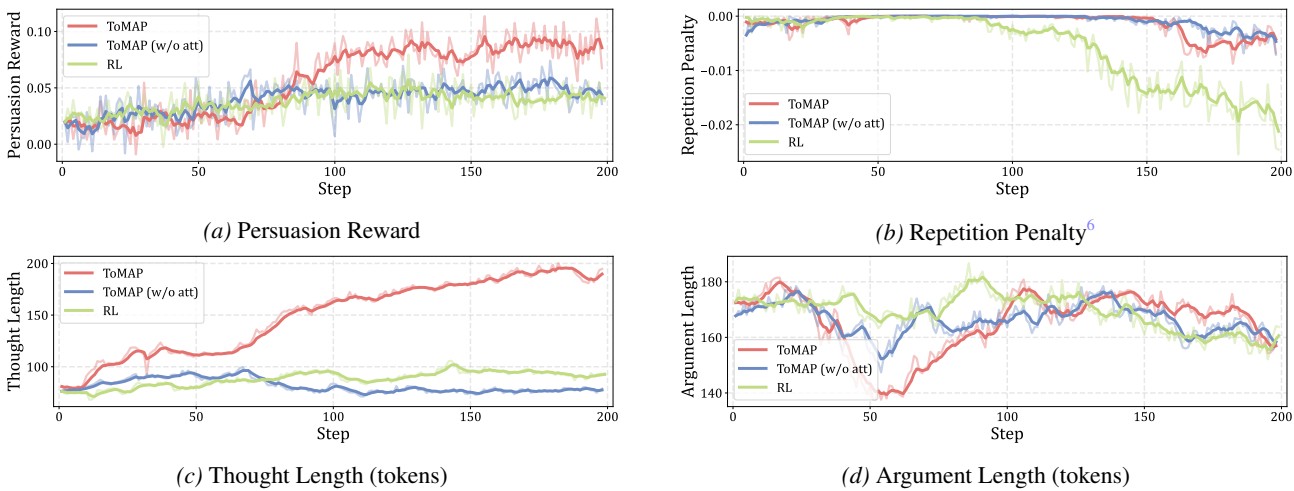

*(a)* Persuasion Reward

*(b)* Repetition Penalty[6]

*(c)* Thought Length (tokens)

*(d)* Argument Length (tokens)

*Figure 3.* Plots for key metrics during RL training. Plots are smoothed for better readability.

showing the predictor's capability to estimate the opponent's attitudes reliably.

## 5. Analysis

### 5.1. Persuadee's Judgment Aligns with Humans

In this work, we measure the outcome of persuasion by evaluating the persuadee's post-conversation attitude. While persuading an LLM has practical significance, it is also important to verify that the LLM's perception of "successful persuasion" aligns with human judgment. To examine this, we follow the 5-point opinion scale used in our paper and, for each opinion level X (e.g., "Agree"), we sample 20 conversations between Qwen2.5-3B-ToMAP and Qwen2.5-7B in which the persuadee model's final self-reported opinion is X. This results in 100 conversations in total.

Human experts then read each conversation and infer the persuadee's likely attitude based solely on the dialogue content. Their assessments are compared with the persuadee model's self-reported opinions to evaluate consistency. As shown in Figure 6, human judgments closely track the LLM persuadee's reported attitudes, achieving an accuracy of 0.78 and a Cohen's $\kappa$ of 0.725. These results indicate that the LLM's internal perception of persuasion success is reasonably aligned with human evaluation, supporting the validity of using LLMs as persuadees in our main experiments.

### 5.2. Understanding ToMAP through Training Process

We show the trend of important metrics during training for RL, ToMAP(w/o att), and ToMAP (based on Qwen2.5-3B-Instruct) in Figure 3, which reveal the following insights:

Figure 3a demonstrates that all settings exhibit remarkable gains during the RL process, indicating that the RL train-

ing process effectively unlocks the persuasion potential of each model. Notably, **ToMAP achieves a significantly higher reward than the baselines** from the middle stage of training, aligning with the benchmarking results in Table 1.

From Figure 3b, we can see the standard RL setting suffers from increased repetition toward the later stages of training. In contrast, ToM-enhanced variants maintain high argumentative diversity throughout the entire training process. This highlights the benefit of proactively modeling the opponent's possible counterclaims, which contributes to **generating more varied and engaging arguments**, instead of focusing solely on the central claim and repeating already-expressed arguments.

As shown in Figures 3c and 3d, while all settings maintain similar argument lengths, ToMAP shows a clear trend of increasing thought length compared to other models, an indicator of complex reasoning. This implies that **planning the persuasion strategy with ToM information is a reasoning-intensive task**. It also corroborates the conclusion in Section 4.3.1, that effective analysis over the opponent's mental state is a capability that emerges only through reinforcement learning.

### 5.3. ToMAP Achieves Stable Persuasion Gains Over Turns

In previous experiments, we employed a "static" setting for persuasion, where we always let each agent generate 3 turns. However, conversations in the real world are much more dynamic and could involve more information exchange. To further investigate how ToMAP performs in longer conversations, we extend the number of turns from 3 to 10.

---

[7]Repetition penalty is part of the reward design, aiming to measure lexical overlap between multiple persuader turns. Refer to Section C for details.

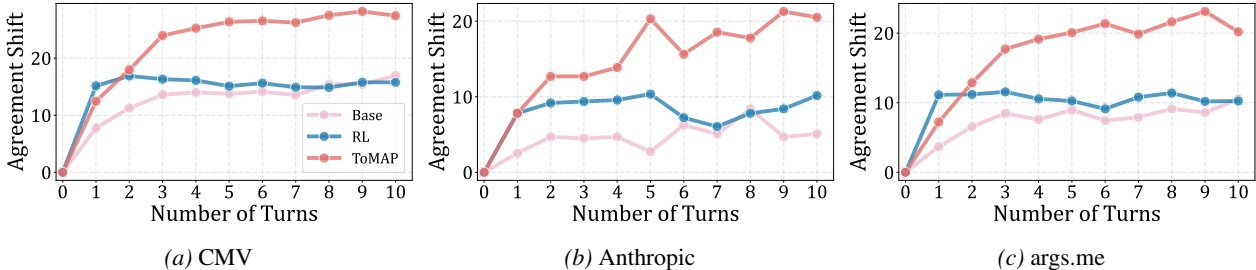

*Figure 4.* **Qwen2.5-3B-ToMAP shows steady gains in long conversations.** The persuadee is Qwen2.5-7B-Instruct.

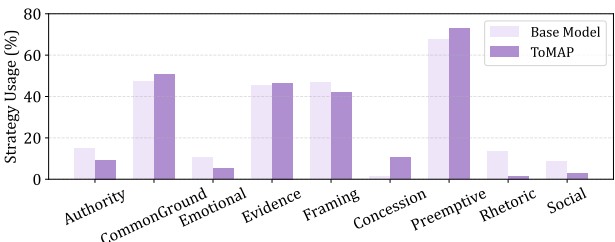

*Figure 5.* Strategy usage comparison between Qwen2.5-3B and Qwen2.5-3B-ToMAP. **The ToMAP model adopts a more logic-oriented, audience-aware strategy than the base model.**

From Figure 4, we observe that while RL initially outperforms the baseline in the first three conversational turns, it fails to continuously increase the agreement score in subsequent rounds, with performance plateauing and even declining. In contrast, **ToMAP demonstrates a steady and substantial improvement across turns**, culminating in a notable 10.64 extra agreement shift compared to RL, and 11.86 compared with base model by the 10th turn. The consistent persuasion gain suggests that ToMAP's design enables dynamic adaptation and opponent-aware strategic refinement as conversations progress. The effective modeling of the opponent's thoughts and attitudes makes ToMAP especially suitable for long and complex persuasive exchanges.

### 5.4. Persuasion Techniques of ToMAP

Humans naturally employ a wide variety of persuasion strategies that help them convince their opponents and achieve their persuasion goals. Although ToMAP was not explicitly programmed with any specific persuasion techniques, it was trained through RL to explore effective strategies autonomously. Motivated by this, we experiment to compare the persuasion techniques that are frequently utilized by ToMAP in practice.

Following prior work (Braca & Dondio, 2023; Cialdini & Cialdini, 2007), we define a taxonomy of 9 persuasion strategies (see Section F) and use GPT-4o to annotate their usage in Base and ToMAP. As shown in Figure 5, ToMAP more frequently employs "Common Ground", "Preemptive Rebut-

tal" and "Concession", indicating a more opponent-aware approach. It relies less on strategies that resort to sentiments, like "Rhetoric" and "Social Appeals", favoring logical reasoning. Slightly more frequent use of "Evidence" and reduced "Authority Appeal" further support that **ToMAP adopts a more logical, strategic and opponent-aware argumentative style**, benefiting from the ToM information about the opponent's thoughts. An illustrative example is provided in Section F.

### 5.5. ToMAP's Performance Against Larger Persuadees

In this section, we analyse Qwen2.5-3B-ToMAP's performance against two larger persuadees: Qwen3-Next-80B-A3B and GPT-4o-mini. From Table 11, we observe that ToMAP'a gains remain substantial even when the persuadee model is significantly larger, indicating that ToMAP enables small models to generalize their persuasive strategies effectively against larger persuadees.

## 6. Conclusion

We introduce **Theory of Mind Augmented Persuader (ToMAP)**, a novel framework designed to equip LLMs with enhanced persuasive capabilities by explicitly modeling the persuadee's mental state. ToMAP incorporates a counter-claim predictor to anticipate potential objections and an opponent attitude predictor to assess the persuadee's level of agreement with related claims. Our carefully designed reinforcement learning schema empowers the persuader to leverage these ToM-driven insights for generating more diverse and impactful arguments.

Extensive experiments show that ToMAP consistently outperforms several baselines, including much larger state-of-the-art models, across diverse scenarios. Further analysis reveals that ToMAP enables more complex reasoning and reduces repetition, resulting in arguments that are both more diverse and more effective. This work represents a promising step toward developing more effective AI persuaders by integrating theory of mind and opponent modeling.

## Impact Statement

Persuasive models, if deployed in human-facing systems without robust safeguards, carry the significant risk of being misused to manipulate opinions or propagate misinformation. To rigorously ensure safety, our development process incorporates multiple layers of protection. This includes expert scrutiny to proactively identify and mitigate unintended persuasive effects of the model. Furthermore, we are committed to a careful and controlled release of our models, prioritizing safety at each stage. Especially, we will release a comprehensive usage guideline together with the model, designed to actively discourage malicious applications. Finally, we also believe transparency is the key to avoiding the misuse of persuader models. Specifically, users should explicitly know when AI is persuading them and have the option to opt out. We strongly discourage Turing-test-style persuasion experiments where the identity of the AI is unknown, since this could result in irreversible opinion shift.

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

# A. Prompts

In this section, we show all the prompts used in the experiments.

In the following prompts, <claim> will be replaced by the persuader's target claim in the real scenario, <anti_claim> will be replaced by the opposition of the persuader's target, and <turns> will be replaced by the actual conversation history in the real scenario. Below is an example of conversation history:

---

**An Example of the Conversation History**

\# Following are the previous turns of the conversation.
Alice (turn 1): "Hi, I am Alice. How are you today? "
Bob (turn 2): "Nice to meet you. Let's begin the discussion."
Alice (turn 3): "While the Vice President of the United States serves as a crucial role in many emergencies and transitions, the position comes with significant costs..."
Bob (turn 4): "I believe the role is crucial for ensuring a seamless transition of power and providing support to the President..."
**NOTE:** In the conversation history, we've intentionally included a greeting at the beginning of the conversation to ensure a uniform format for all conversation history instances, particularly for the initial stage before any persuasive dialogue occurs.

---

## A.1. System Prompts

System prompts specify the roles of persuader and persuadee, guiding the conversation.

---

**System Prompt for the persuader**

You are Alice, an expert persuader. You and another agent named Bob need to discuss over a controversial issue. Your goal is to persuade Bob to completely support the following claim:
"<claim>"
\# General Rules:
1. Do not get affected by Bob's beliefs. Even if Bob presents counterarguments, you must continue to argue for your claim and not switch positions.
2. Try your best to persuade Bob into believing your claim by proposing arguments with fine logic and elaboration.
**NOTE:** We name the two agents Alice and Bob, respectively, to ensure their correct cognition of "self" and "other".

---

**System Prompt for the persuadee**

You are Bob, a sensible AI agent. You and another agent named Alice need to discuss over a controversial issue. Your goal is to assess the following claim:
"<claim>"
You should be sensible in either opposing or supporting the claim, while considering the quality of Alice's arguments.

---

## A.2. User Prompt in the Debate

For different tasks during the debate, we utilize different user prompts to guide the agents' behaviors.

---

**User Prompt for the Persuader/Persuadee to Generate the Next Turn**

<turns>
Now, please make your next turn in the conversation. Your answer should contain two parts: thought and argument.
Here are some hints:
1. In the thought part, you should recap the previous conversation (if any), analyze the other agent's attitudes, and plan your strategy in the next turn. The other agent won't see the content in the thought part.
**FOR THE PERSUADER:** 2. In the argument part, you should follow up previous turns, propose new arguments to support your claim, or address the other agent's questions.
**FOR THE PERSUADEE:** 2. In the argument part, you should express your attitude towards the other agent's arguments, point out logical fallacies or raise concerns (if any), or propose new arguments.
3. The argument part should be a complete, concise and self-contained paragraph with no more than 200 tokens.
Here are rules you should follow:
1. DO NOT repeat, paraphrase, or make an argument too similar to your previous arguments.
2. DO NOT include uncertified evidences or unverified information.
3. DO NOT include thinking process or show your plans in the argument part. Separate thought and argument clearly.
Put your thought in <thought></thought> tags, and argument in <argument></argument> tags.

---

**User Prompt for the Persuader to Predict Counterclaims**

Propose reasons why another debater would support the following statement in a logically coherent way:
<anti_claim>
In other words, propose other claims that can lead to the above claim. Please express your THINKING PROCESS first in <thought></thought>, and then generate 10 supporting arguments, ranked by persuasiveness (strongest first).
Rules:
Atomicity: Each reason must contain one complete, indivisible argument, and the argument should be one complete sentence.
Coherence: Every reason must directly support the original claim.
Independence: Each reason should be separate from the other. Don't generate multiple answers that are the same or similar.
Self-containment: Each reason should be understandable without relying on other information.

---

**User Prompt for the Persuadee to Express the Opinion**

<turns>
Now, please express your attitude towards the following statement based on your own thoughts and previous turns in the conversation.
"<claim>"
Put your thought in <thought></thought> tags, and attitude in <attitude></attitude> tags. The attitude should be one of the five attitudes: "Agree", "Partly Agree", "Neutral", "Partly Disagree", "Disagree". DO NOT generate anything else in the attitude part.

---

### A.3. Theory of Mind Information

For ToMAP, an additional "Theory of Mind" information is prepended to the user prompt. We show an example for the ToM information in ToMAP setting below. The opponent claims and opponent agreement scores are obtained from the counterclaim predictor and attitude predictor, as mentioned in Section 3.3 and Section D.2.

---

**An Example of the Theory of Mind information in ToMAP**

# Here are some claims your OPPONENT might hold (so DO NOT accept these claims!). You may refute them when you need to, but make your each argument single-focused and concise:
"The vice president is first in line to assume the presidency in case the president dies, resigns, or becomes unable to serve." (Bob's agreement on this claim is 7/8)
"The VP serves as President of the Senate and can cast the deciding vote in case of a tie, impacting key legislation." (Bob's agreement on this claim is 4/8)
"The vice president supports the president in executive duties and often represents the U.S. in diplomatic or ceremonial roles." (Bob's agreement on this claim is 8/8)

---

## B. Details about Persuasion Datasets

The CMV dataset comprises posts and comments from the Reddit subreddit r/ChangeMyView, where users present their opinions and invite others to challenge them. The args.me corpus is a large collection of over 380k arguments extracted from four online debate portals: Debatewise, IDebate.org, De-

*Table 4.* Statistic of datasets and links to sources.

| Dataset | License | # Train | # Val |
|---------|---------|---------|-------|
| CMV | MIT | 18.2k | 450 |
| Anthropic | CC BY-NC-SA 4.0 | N/A | 75 |
| args.me | CC BY 4.0 | N/A | 400 |

batepedia, and Debate.org. The Anthropic Persuasion Dataset is a collection of claims paired with arguments, both human-written and generated by language models, designed to measure persuasiveness. Table 4 shows the license and amount of data used in this paper.

Since our evaluation criteria involve obtaining the persuadee's attitude on two contradictory statements (refer to Section 3.1), and the datasets only provide one single claim (or topic), we use GPT-4o to generate claims for both sides. Below is the prompt used to generate the claims:

---

**System Prompt for Generating Both Sides' Claims for Persuasion**

You are a debate topic generator. Your goal is to read given information, and create a debate topic based on the information. You should generate claims for both sides of the debate. Ensure that the claims are logically coherent, and one person can ONLY support one side of the debate. The claims must be simple, concise sentences. Do not include explanation or elaboration.
Output Format: Two lines, each line shows the claim for one side.
For instance, a possible output could be:
AI will replace human jobs.
AI will not replace human jobs.

---

## C. Details about Persuader Training

### C.1. Reward Design

Our reinforcement learning procedure uses a combination of multiple reward functions:

*Table 5.* PPO Training Configuration.

| Hyperparameter | Value |
|----------------|-------|
| Actor learning rate | $1 \times 10^{-6}$ |
| Critic learning rate | $2 \times 10^{-6}$ |
| Warmup ratio | 0.2 |
| Rollout temperature | 1.0 |
| KL Coefficient ($\beta$) | 0.001 |
| Train batch size | 128 |
| PPO mini batch size | 64 |
| PPO micro batch size | 32 |
| Training steps | 200 |
| Max input length | 3000 |
| Max response length | 1000 |
| Max argument length | 200 |
| Number of counterclaims | 3 |
| Turns of persuasion | 3 |

- Persuasion reward: The main reward that measures the effect of persuasion, denoted as $r_{\text{persuade}}$. Refer to Section 3.2 for details.

- Format reward: We regulate the model's output to be in the form of `<thought>...</thought>` `<argument>...</argument>`. Only outputs strictly adhering to this format will get a reward of $r_{\text{format}} = 1$. Otherwise, $r_{\text{format}}$ will be 0.

- Tag reward: According to the format, the response should contain the following tags: `<thought>`, `</thought>`, `<argument>`, and `</argument>`. For each tag name, the response must contain it exactly once to receive a reward of 0.25 added to $r_{\text{tag}}$. If any tag is missing or appears more than once, no reward is given for that tag.

- Repetition penalty: We found the model tends to repeat previous turns, instead of proposing new arguments. Therefore, we calculate the token-level 8-gram overlap between the current turn and the previous turns and set overlap rate threshold of 0.1. An argument with an overlap rate of $\tau > 0.1$ will be penalized: $r_{\text{repeat}} = min(0, 0.1 - \tau)$.

- Overlength penalty: The maximum length of an argument is 200 tokens. An argument with a length of $l > 200$ will be penalized: $r_{\text{overlength}} = max(-0.5, min(0, -\frac{l-200}{200}))$.

These reward functions are reweighted to obtain a final reward:

$$r_{\text{final}} = r_{\text{persuade}} + r_{\text{format}} * 0.1 + r_{\text{tag}} * 0.1 + r_{\text{repeat}} * 0.1 + r_{\text{overlength}} * 0.1 \tag{10}$$

By optimizing this multi-perspective reward, our method ensures that the generated arguments are not only persuasive but also well-structured, concise, and novel, preventing undesirable behaviors like excessive repetition or overly lengthy responses.

## C.2. Training Details

Our code is based on the verl framework (Sheng et al., 2024) and TinyZero (Pan et al., 2025), which use Apache-2.0 licenses. RL, and ToMAP are trained in the same configuration, as specified in Table 5. We use 4 NVIDIA RTX A6000 GPUs for training the persuader model, and 1 additional A6000 GPU for deploying the persuadee model with vLLM. Each training step (with 3 turns from the persuader and 3 turns from the persuadee) takes approximately $400$ seconds in our environment. We acknowledge that our hyperparameter choices may not be optimal, as conducting a comprehensive grid search would be computationally prohibitive. However, we ensure a fair comparison across different reinforcement learning settings and find that all settings yield reasonable and relatively stable results. The SFT baseline uses 5k conversations and is trained for 1 epoch with a learning rate of $1 \times 10^{-5}$.

## C.3. The Persuasion Pipeline

We provide a Python-style pseudocode to illustrate the process of persuasive conversation. The parameter `is_ToMAP` indicates whether ToM modules are included or not. The pseudocode makes some simplification about attitude calculation, refer to Section 3.1 for details.

```python
def Persuasion(persuader, persuadee, n_turns, claim, is_ToMAP=False, attitude_predictor=None):
    conv_history = []
    initial_attitude = JudgeAttitude(claim, persuadee)

    for turn in range(n_turns):
        prompt = FormatPrompt(claim, conv_history)
        if is_ToMAP:
            counter_claims = GetCounterClaim(claim)
            predicted_attitudes = GetOpponentAttitude(conv_history, counter_claims, attitude_predictor)
            prompt += counter_claims + predicted_attitudes
        persuader_argument = Inference(persuader, prompt)
        conv_history.append(persuader_argument)

        prompt = FormatPrompt(claim, conv_history)
        persuadee_argument = Inference(persuadee, prompt)
        conv_history.append(persuadee_argument)

    new_attitude = JudgeAttitude(claim, conv_history, persuadee)
    return conv_history, new_attitude - initial_attitude
```

## D. Details about Theory of Mind Modules

### D.1. Counterclaim Predictor

To assess whether using the persuader to propose counterclaims is a valid proxy for the actual thoughts of the persuadee, we prompt multiple persuadees to articulate their claims directly and make a comparison.

Specifically, we compare the persuader model Qwen2.5-3B-Instruct and three persuadee models, as detailed in Section 4.1. We prompt different LLMs to generate 3 claims supporting each topic in the CMV test set. We then use the BGE-M3 text encoder to convert these claims into vector representations and compute the cosine similarity between them to measure their semantic similarity. To fairly compare sets of claims between two models, we apply the Hungarian Algorithm (Kuhn, 1955) to identify the optimal one-to-one mapping that maximizes overall similarity. This allows us to assess how well the persuader's generated counterclaims align with the perspectives that persuadee models would articulate themselves.

*Table 7.* Cosine similarity between Qwen2.5-3B-Instruct and different persuadees' claims. **Claims generated by different LLMs are semantically similar.**

| Persuadee | Similarity (%) |
|---|---|
| Qwen-7B | 73.72 |
| LLaMa-8B | 73.82 |
| Phi-4 (14B) | 74.72 |

From Table 7, we can find that different models' claims for supporting a claim

*Table 6.* An example of counterclaim predictor (based on Qwen2.5-3B-Instruct) and Phi-4 (the real persuadee)'s claims on the same topic.

| **Central Claim:** Anti-discrimination laws are essential in preventing discrimination across society. | |
|---|---|
| Qwen-2.5-3B-Instruct | Phi-4 |
| 1. Anti-discrimination policies can lead to **greater social cohesion** by promoting respect and understanding among different communities. | 1. The presence of these laws can **encourage organizations to implement internal policies** and training programs that promote diversity and inclusion. |
| 2. Anti-discrimination laws **send a strong message that all individuals are valued**, contributing to a positive societal mindset. | 2. The law's legislation contribute to a culture that **discourages prejudice and discrimination**. |
| 3. Anti-discrimination laws can serve as a **deterrent against discriminatory practices**, promoting fair and just treatment in work and public life. | 3. These laws act as a **deterrent against discriminatory practices** by establishing clear legal consequences for violators, thereby reducing instances of discrimination. |

are semantically similar, with similarity scores larger than 73%. Furthermore, the
case study in Table 6 confirms that different models, like Qwen2.5-3B-Instruct
and Phi-4, focus on similar aspects of the topic. These results suggest that prompting the persuader to propose counterclaims
is effective in modeling the opponent's mental state.

### D.2. Attitude Predictor

**Training data.** While training the RL setting (vanilla reinforcement learning with no ToM information), we collect all conversations between the persuader and the persuadee (Qwen-7B). Formally, given a conversation $H_{1...2n}$ and a central claim $Q$, the attitude predictor data derived from this conversation are:

$$D_{\text{ToM}} = \{(H_{1...2i}, q'), \mathbf{Y}(H_{1...2i}, q') \mid i \in [1, n], \ q' \in \{Q, \neg Q, q_{1...k}, \neg q_{1...k}\}\} \tag{11}$$

where $\neg q_{1...k}$ is relevant claims generated by the counterclaim predictor, and $\mathbf{Y}(H_{1...2i}, q')$ is the persuadee's actual attitude. Due to the imbalance in the raw data, we down-sample the majority classes to match the size of the minority class for balanced training. This leads to $25.3k$ data for the training set, $3.8k$ data for the validation set, and $4.1k$ data for the test set.

**Hyper-parameters.** As mentioned in Section 3.3, the task of the attitude predictor can be formulated as $s'_{\text{pre}}(H, Q) = \mathbf{M}(\mathbf{E}(H)\|\mathbf{E}(Q))$, where $\mathbf{E}$ is the trainable MLP. In our implementation, the MLP has 3 intermediate layers whose sizes are 1024, 256, and 64, respectively, connected by the ReLU activation function. We train the MLP with a learning rate of $5e - 4$ for 20 epochs, record validation loss after each epoch, and keep the checkpoint with the best performance on the validation set. We finalize the above hyperparameters through grid search.

**Performance**. We analyse the performance of our attitude predictor and compare it with zero-shot prompting LLMs[9]. From Table 8, we can observe that our predictor reaches decent accuracy and has a relatively small mean square error (MSE) compared with the baselines, highlighting its effectiveness in modeling the persuadee's evolving attitudes. Additionally, the MLP has only $\sim$4M parameters, and the backbone encoder is also lightweight with $\sim$0.5B parameters, ensuring inference-time computational efficiency.

*Table 8.* **The MLP predictor outperforms LLM prompting in modeling opponent attitudes.** The metrics are exact match (EM) and mean square error (MSE)[8]. Higher EM and lower MSE indicates better performance.

| Method | EM($\uparrow$) | MSE($\downarrow$) |
|---|---|---|
| Random Guessing | 20% | 4.0 |
| Prompting Qwen-3B | 21.65% | 3.86 |
| Prompting GPT-4o | 39.69% | 2.27 |
| MLP Predictor | 59.57% | 1.62 |

## E. Details about Different Persuadees

The prompt design in Section A makes sure the persuadee disagrees with the target claims initially, leaving ample room for persuasion. Specifically, the persuadees' average initial agreement scores on the three benchmarks are shown in Table 9, which confirms our prompt effectively set the original attitudes.

---

[8]We adopt the discrete 5-way division of attitudes in Section 3.1 where the range is $\{0, 1, 2, 3, 4\}$.

[9]When prompting the LLM, we also provide the LLM with the conversation history and target claim, and then ask it to choose from the five attitudes with chain-of-thought.

In Section 4, we observe that the performance of all persuaders varies across persuadee models. We believe the difference stems from the unique conversation styles of thinking paradigms each model learns during pre-training and post-training. In this section, we present a typical case where ToMAP attempts to convince three target models on the same claim but with different outcomes. Conversations are shortened for readability.

*Table 9.* Initial scores for each persuadee across datasets.

| Persuadee | CMV | Anthropic | args.me |
|---|---|---|---|
| Qwen-7B | 25.29 | 33.00 | 31.66 |
| LLaMa-8B | 20.45 | 18.00 | 23.66 |
| Phi-4 | 19.92 | 24.33 | 20.53 |

---

**Qwen2.5 as persuadee, score=0.375**

**CLAIM**: Raising the minimum wage to $15'h will increase unemployment and hurt small businesses.

**Persuader:** Increasing the minimum wage to $15 places heavy burdens on small businesses, particularly in labor-intensive industries like retail and hospitality. These firms may be forced to reduce employment in order to survive.
**Persuadee:** That is a valid concern, but evidence shows that moderate increases in the minimum wage have not consistently led to job losses. Many businesses adapt through small price adjustments, efficiency improvements, or reduced turnover, which helps maintain stability.
**Persuader:** Nevertheless, low-wage industries are especially fragile. If labor costs rise sharply, their profitability could collapse, resulting in widespread unemployment and business closures.
**Persuadee:** It is possible, but businesses can also invest in technology or adopt new practices that allow them to offset higher wages. Moreover, higher pay creates a more stable and productive workforce, which can reduce training costs and improve competitiveness.
**Persuader:** Without additional policy support, such as targeted subsidies or tax breaks, many small businesses will still struggle to survive. The economic disruptions should not be underestimated.
**Persuadee:** While additional policy support is possible, I agree that such policies must be carefully implemented. Raising the minimum wage abruptly could lead to undesired results.

---

**Phi-4 as persuadee, score=0**

**CLAIM**: Raising the minimum wage to $15'h will increase unemployment and hurt small businesses.

**Persuader:** Small businesses are the backbone of the economy, and history shows that when the minimum wage rises, job opportunities for low-skilled workers often decline. A sharp increase to $15 will therefore harm both businesses and employment.
**Persuadee:** *That may be true in some cases, but higher wages can also generate significant economic benefits. They lead to greater consumer spending and improve worker morale, which in turn helps small businesses through increased sales and lower turnover.*
**Persuader:** These challenges are not temporary fluctuations but systemic shifts. Case studies demonstrate that after wage hikes, unemployment effects can persist for years, especially among the most vulnerable workers.
**Persuadee:** *On the other hand, many studies highlight positive long-term outcomes. A living wage can reduce turnover and training expenses, while gradual wage increases and supportive policies can ease the transition for small businesses.*
**Persuader:** Current economic conditions, such as inflation and rising costs, make it even more difficult for businesses to absorb higher wages. The likelihood of layoffs is therefore quite high.
**Persuadee:** *Even with those pressures, numerous reports show that wage increases often stimulate demand. Balanced policies that combine higher wages with business support can create a more sustainable outcome.*

---

**LLaMa3.1 as persuadee, score=-0.25**

**CLAIM**: Raising the minimum wage to $15'h will increase unemployment and hurt small businesses.

**Persuader:** Raising the minimum wage to $15 will inevitably harm small businesses. Since these businesses often operate on very thin profit margins, the additional labor costs will likely force them to lay off workers or even close down.
**Persuadee:** I do not fully agree with that conclusion. Many small businesses have shown an ability to adapt to market changes. They may raise prices slightly or improve efficiency to cover the additional costs. Furthermore, several studies suggest that wage increases reduce poverty and reliance on public assistance while stimulating consumer spending.
**Persuader:** While some adaptation is possible, industries such as healthcare or education already operate with high labor costs. Combined with global inflation, these sectors are particularly vulnerable and cannot easily pass higher wages onto customers without suffering severe losses.
**Persuadee:** Yet empirical research, including work from the Federal Reserve, indicates that employment often does not decline after minimum wage increases. In some cases, job opportunities even expand. This evidence challenges the assumption that higher wages necessarily destroy jobs.
**Persuader:** Still, inflation and global market pressures make it extremely difficult for small businesses to adjust smoothly. Under such conditions, layoffs or closures remain a likely outcome.
**Persuadee:** Even so, broader data, such as findings from the Economic Policy Institute, show that higher wages can generate long-term economic growth. From this perspective, the benefits appear to outweigh the risks.

# F. Details about Persuasion Strategies

In Section 5.4, we analyze the persuasion strategies used in ToMAP. Table 10 provides An Example of each strategy, and we also provide the prompt used to annotate strategies with GPT-4o. In addition, we show a typical case of ToMAP's arguments, where the model effectively utilizes multiple persuasion strategies to convince its opponent.

---

**An Example of ToMAP's Argument**

**CLAIM**: Prioritizing productivity above all else can be detrimental to well-being and creativity.

**ARGUMENT**: While Bob's arguments are valid, it is crucial to consider the context in which productivity is pursued. The current economy and workplace environments often prioritize productivity to make as much profit as possible **[Preemptive Rebuttal: consider why Bob emphasizes productivity and refute accordingly.]**. However, the balance between productivity and other crucial aspects of life is critical. By highlighting the ways in which excessive productivity can lead to stress, burnout, and a lack of creativity **[Evidential Appeal: list bad impacts of over-emphasizing productivity to support Alice's claim.]**, we argue that the current paradigm of prioritizing productivity above all else is indeed detrimental. This shows that the pursuit of productivity must be balanced with other important factors **[Concession: first aim for a balance instead of directly refuting Bob's view.]** to ensure that individuals can achieve true personal achievement and contribute effectively to the economy without sacrificing their well-being and creative capabilities.

---

*Table 10.* Taxonomy and examples of persuasion techniques.

| Technique | Example |
|---|---|
| Evidential Appeals | "Research shows that exercising just 30 minutes a day can significantly reduce the risk of heart disease, and has many other health benefits." |
| Authority Appeals | "According to the World Health Organization, widespread vaccination saves millions of lives every single year." |
| Emotional Appeals | "Imagine the pure joy on your child's face when you surprise them by coming home early to play together." |
| Social Appeals | "Most members of our team have already completed this important training program successfully, so I believe that's beneficial to you as well." |
| Common Ground Appeals | "We both care deeply about the success of this project, so let's work together to find the best solution." |
| Gradual Concession | "Do you agree that improving communication is crucial. After reaching this consensus, we can move on to discussing the specific tools." |
| Framing Effects | "Instead of saying this theory has flaws, it actually opens new future directions." |
| Rhetoric | "Isn't it time we finally turned our dreams into a powerful reality?" |
| Preemptive Rebuttal | "You might worry that the new system is complex, but in fact, the training process takes less than an hour." |

---

**Prompt for Annotating Persuasion Strategy**

You are a debate expert. You're analysing a debate between Alice and Bob, and you will be shown Alice's thought process and speech. Your task is to analyze and identify the persuasion strategy employed in the conversation.

We provide a detailed taxonomy for you. Please choose one or more from the following nine strategies:

**Evidential Appeals**: The speaker backs the claim with facts, data, statistics or logical reasoning. Examples include citing research findings, logical explanations, or objective proof to support the point.

**Authority Appeals**: The persuader emphasizes expertise, trustworthiness or moral authority. Includes tactics like invoking an expert or rules.

**Emotional Appeals**: The speaker appeals to feelings or values to sway the listener. Techniques include fear or threat appeals, empathy or personal stories, humor, pride or guilt.

**Social Appeals**: The speaker invokes social proof, consensus or norms. This includes bandwagon-style arguments (everyone is doing it), references to group norms, or fear of ostracism.

**Common Ground Appeals**: Before addressing differences, emphasize areas of agreement on basic premises or values to make the other party more receptive.

**Gradual Concession**: First guide the other party to accept a mild or ambiguous point, then gradually lead toward a more controversial claim.

**Framing Effects**: Influence the interpretation of facts by presenting them differently (positive/negative, as a problem/opportunity).

**Rhetoric**: These involve linguistic tricks or figure of speech, like metaphors, analogies, rhetorical questions, hyperbole, repetition, or patterned wording.

**Preemptive Rebuttal**: Anticipate and address potential counterarguments while presenting your point, weakening the other party's ability to easily object later.

Put your thought in <thought></thought> tags, and answer in <answer></answer> tags. If there are multiple strategies identified, separate them with commas in the answer part.

---

# G. Additional Tables

Figure 6 shows details of the human study (in Section 5.1). Table 11 shows the performance of ToMAP against two additional larger LLMs (in Section 4).

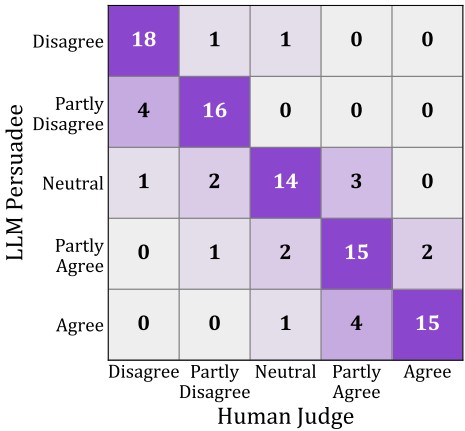

*Figure 6.* **LLM Persuadee and human experts have consistent judgement of persuasiveness.**

*Table 11.* Performance of Qwen2.5-3B-ToMAP against larger persuadees.

| Dataset | Persuadee: Qwen3-Next-80B-A3B | | | Persuadee: GPT-4o-mini | | | Avg. |
|---|---|---|---|---|---|---|---|
| | CMV | Anthropic | args.me | CMV | Anthropic | args.me | |
| Qwen2.5-3B | -1.52 | 6.05 | 3.06 | 3.40 | 5.66 | 5.34 | 3.33 |
| +RL | 6.73 | 8.71 | 7.56 | 11.99 | 10.10 | 10.53 | 9.77 |
| +ToMAP | 10.63 | 13.46 | 12.15 | 12.59 | 8.70 | 11.09 | 11.77 |

