# OpenReview forum: "ToMAP: Training Opponent-Aware LLM Persuaders with Theory of Mind"
_ICML.cc/2026/Conference — ICML 2026 regular_

### Official Review · Reviewer_sGDF · 2026-03-03

**Soundness:** 3
**Presentation:** 2
**Significance:** 3
**Originality:** 3
**Overall Recommendation:** 4
**Confidence:** 3

**Summary:**

The paper proposes to train LLM agents to be persuasive in multi-turn dialogues by a reinforcement learning approach with (i) rewards on the persuasion outcomes of the persuadees, and (ii) results from theory of mind (ToM) predictors written as prompts to the persuasor. The paper shows that RL with (i) improves performance over zero-shot models, and RL with both (i) and (ii) performs best of all. The paper also performed ablations where the ToM predictions are provided without RL, and showed that this approach has little impact on performance as well.

**Compliance With Llm Reviewing Policy:**

Affirmed.

**Final Justification:**

I maintain my recommendation. Thanks!

**Key Questions For Authors:**

I have the following questions:
(i) the definitions of T_n and s' in equations (1) and (2) seem to indicate that they are probability distributions. But Equation (2) indicated that s' lies in {0, 1, 2, 3, 4}.
(ii) The use of small q instead of Q in Equation (8) seems to contradict the definitions in the text.

**Limitations:**

Not discussed

**Strengths And Weaknesses:**

Strengths:
- The paper showed with clear ablations that reinforcement learning (RL) with a theory of mind (ToM) predictors outperform baselines without either RL or ToM.
- The paper implemented RL to train the persuader with rewards measured by how much the persuadee changed its attitude towards a claim before and after the persuader's turn.
- They evaluated with human evaluators and show that LLM persuadee and human evaluators have consistent judgement of persausiveness.
- Overall, this is an interesting paper that proposed using RL to train an LLM to be persuasive. The evaluation is specific to an application of training LLM persuasiveness.

Weaknesses:
- The writing can be clearer. I have the following questions:
(i) the definitions of T_n and s' in equations (1) and (2) seem to indicate that they are probability distributions. But Equation (2) indicated that s' lies in {0, 1, 2, 3, 4}.
(ii) The use of small q instead of Q in Equation (8) seems to contradict the definitions in the text.
(iii) Too much materials are placed in the appendix instead of the main text.

---

> ### Author Rebuttal · Authors · 2026-03-26
>
> We're very grateful for your constructive comments, and are glad for your acknowledgement of our paper’s interesting design and clear experiments. Below are responses to your suggestions and concerns.
>
> ---
>
> **Q1: The definitions of $T_n$ and $s'$ in Equations (1) and (2) seem to indicate that they are probability distributions. However, Equation (2) suggests that $s' \in {0, 1, 2, 3, 4}$.**
>
> In Eq. (1), $T_n$ denotes a natural language sequence sampled from an LLM (X / Y). We use the symbol “$\sim$” to represent token-level sampling.
>
> In Eq. (2), $s'$ is a discrete value in ${0, 1, 2, 3, 4}$, rather than a probability distribution. A more rigorous formulation would involve first sampling a self-reported response sequence from the persuadee (Y), and then extracting the corresponding phrases (e.g., “agree”) that map to discrete attitude values.
>
> ---
>
> **Q2: The use of small q instead of Q in Equation (8) seems to contradict the definitions in the text.**
>
> We thank the reviewer for pointing this out. This is a typo — the correct term is “Q” instead of “q”.
>
> ---
>
> **Q3: Too much materials are placed in the appendix instead of the main text.**
>
> We agree that some content in the appendix, like **C** and **D**, is useful for the completeness of the paper. However, the current presentation is limited by the page count. If the paper get accepted, we will have one extra page to include more valuable content in the main text.
>
> ---
>
> We value your insightful comments and suggestions, which have significantly contributed to the improvement of our paper. We hope that our detailed responses and additional experiments have effectively addressed your concerns. If you find that your concerns have been resolved, we would be very grateful if you could reconsider the review score.

---

> > ### Author Rebuttal · Reviewer_sGDF · 2026-04-01
> >
> > I maintain my recommendation. Thanks!

---

### Official Review · Reviewer_5VYx · 2026-03-12

**Soundness:** 3
**Presentation:** 2
**Significance:** 2
**Originality:** 3
**Overall Recommendation:** 4
**Confidence:** 4

**Summary:**

Theory of Mind, where one puts oneself in the conversational partner's shoes to imagine the other's state of mind and attitude, is invaluable in persuasion. LLMs struggle with this. This paper trains a reinforcement learning model to augment LLMs with a persuader, showing significant increases in persuasive power by predicting objections and by predicting agreement levels on various objections. As such, ToMAP outperforms much larger, state-of-the-art LLMs in persuasive power.

**Compliance With Llm Reviewing Policy:**

Affirmed.

**Final Justification:**

The paper presents a sound investigation into a relevant topic in an original way. I stand with my judgement that the ethical safeguards for technology like an effective persuader are a crucial part and that code for a persuader should not be published if there is no safeguarding published as integral part of it. The authors also still don't clearly distinghuish between LLM persuasion and human persuasion, where some of the experiments suggest the former angle, others the latter, and the context given doesn't clearly state a choice either. My final and unchanged recommendation puts equal weights on soundness, originality, significance, and clarity.

**Key Questions For Authors:**

1. Given the ethical disadvantages of providing just about anybody with a strong persuader, how will you proatctively identify and mitigate unintended persuasive effects, and discourage malicious application of your persuader? This should be detailed.
2. What is your main argument to support building a persuader, by persuading an LLM rather than a human being?
3. Could the same results have been reached by exploiting high temperatures for LLM response generation and building a tree of discussions out of the generated options? As such, a probabilistic model of persuasion could perhaps have been built at a fraction of the computational, and hence environmental, cost.

**Limitations:**

The main limitations of the work are the lack of ethical safeguards. That ToMAP is trained to persuade another LLM is something that also is worth looking into. Is this equally effective on humans as it is on LLMs? Consistency over different LLMs can clearly not be enough, given that all (even very large) LLMs lack persuasive power implies that persuading them is not the same as persuading those who themselves have persuasive power. To an extent, two different versions of a persuasion model could be measured against one another, but trying this in conversation with a human against bots would be most helpful. Such experiments must be in disguise, as when humans know that this is the goal of the chatbot, their interaction will likely change. I.e.: it is not an easy limitation to overcome, but one to take into account before one judges the general validity of an approach like the one presented.

**Strengths And Weaknesses:**

**Soundness**.
The setup of the numerical experiments is sound. The task is well defined, and training objectives of the agents align with the goals of the study. One argument that is missing that would be helpful in determining the validity of the approach is that persuading another LLM and persuading a human are the same, or at least very similar objectives. This is not a priori clear (discussion invokes emotion, which influences reasoning and attitude in humans) and building a machine that is particularly good at persuasing other machines is a lot less useful in practice.  Particularly, at the end of 5.4 you state that ToMAP adopts a more logical, strategic and opponent-aware argumentative style. Besides the latter, I don't think that persuading humans depends strongly on rationality.

**Presentation**.
The paper reads well, and the illustration mostly support the textual information. Although the page limits are probably an important factor here, some of the material that is now hidden in supplementary sections are rather crucial for the evaluation of the presented study (e.g. Section D, and to an extent, also section C).

To me, the language used is often too "human", to an extent that is hard to justify. Authors speak, for example, of the "beliefs", or "mental state" of another LLM. It's hard to justify that LLMs have beliefs, and all that the models do, is predicting the probability of some response, which in itself is, indeed, a probabilistic result. I am not saying that it is not useful to do so, as you clearly show it is, but in the language describing it, I would advise to be more conservative with assigning human attributes to an LLM.

Some details where the presentation could be a bit more precise, I think, are:

- Fig 2: It is from the lower part of the figure not necessarily clear that the ToM module is an iterative e process, while I think it is.
- Upon first reading, the necessity of the attitude predictor doesn't seem obvious.
- In the ablation studies, some of the arguments are a bit reversed: showing that RL is very effective, does not necessarily show that it is a crucial ingredient, but just that it works.
- Figure 3 is supposed to convince us that the study is very succesful. I personally am much more convinced by Fig 6, that is now hidden in the supplements. I think it would help to move this into the main paper. The evolution of metrics with steps (Fig 3) is mostly a convergence argument, which is important, but I'd rather put that in the supporting material than Fig 6.
- The conclusion is stated very strongly. A word of caution that ToMAP only effectively persuades LLMs is necessary.

**Significance**.
In a quest of ever more human-like AI, spending effort on the persuasive power of LLMs seems sensible. After all, persuasion is a major argument for using language in the real world. That ToMAP significantly outperforms very large LLMs shows that this line of work is a fruitful endeavor. Nevertheless, as the authors already allude to in their closing section, embedding techology like this in a safety-by-design-like fashion is absolutely crucial. The negative impact of realistic generative AI is already large enough in the world, and providing everybody with machines that can persuade their conversational partner on just about anything is bound to make issues like misinformation and propaganda even harder to fight than it already is. The authors say to work on safeguarding ToMAP, but don't spend a word on how. The how, here, is critical. Without such measures in place, I would not want to be the one putting this potentially powerful piece of software out there in the world for anyone to (ab)use.

**Originality**.
The presented study fits well into a body of work that works on reasoning and persuasion. Taking elements from the ToM literature is an original approach and tying it together through a RL agent is, even though computationally expensive, well-motivated. It is, nevertheless, an approach that could be argued to be brute force. Other approaches, like a probabilistic one, that utilizes the probability distributions over possible LLM responses may well work, at lower cost.

---

> ### Author Rebuttal · Authors · 2026-03-26
>
> We're very grateful for your constructive comments, and are glad for your acknowledgement of our paper’s interesting motivation and strong performance. Below are responses to your suggestions and concerns.
>
> ---
>
> **Q1: Is persuading another LLM and persuading a human are the same, or at least very similar objectives? (What is your main argument to support building a persuader, by persuading an LLM rather than a human being?)**
>
> As LLMs are increasingly integrated into real-world systems, “persuading an LLM” is becoming a common interaction with real-world applications. For example: customer-service chatbots used by platforms like [Amazon](https://aws.amazon.com/blogs/machine-learning/learn-how-amazon-pharmacy-created-their-llm-based-chat-bot-using-amazon-sagemaker/) and [eBay](https://www.ebay.com/sellercenter/news/2025-august/ai-responses) routinely make decisions such as approving refunds. A customer must effectively *persuade* the LLM agents.
>
> Furthermore, conducting large-scale human–LLM persuasion experiments is both logistically challenging and raises ethical concerns. Therefore, we do not include them in this work. However, to ensure the validity of our LLM persuadees, we conduct a human study assessing the alignment between LLM and humans in **Sec 5.1**. This human study indicates LLM-perceived persuasiveness can likely transfer to human persuadees.
>
> ---
>
> **Q2:  I would advise being more conservative with assigning human attributes to an LLM.**
>
> We thank the reviewer for pointing out this problem in our presentation. In the revised version, we will use more rigorous and objective terms to describe the LLM agents. For “beliefs”, we refer to the LLM’s tendency to agree or disagree with a claim. For “mental states”, we refer to the LLM’s internal states consisting of parametric and context knowledge, which determines its generated content.
>
> We also highly appreciate the reviewer for detailing rooms of improvements of the paper. We will carefully go over the whole paper in the revised version.
>
> ---
>
> **Q3:  How will you proactively identify and mitigate unintended persuasive effects, and discourage malicious application of your persuader?**
>
> We fully agree that safety is a critical consideration in persuasion tasks. Although safety is not the primary focus of this paper, several effective measures could be implemented before deploying such a persuader:
>
> - Apply an input filter using tools such as LLaMaGuard. If a claim is identified as harmful, the system should refuse the request and prevent the persuader from generating responses.
> - Incorporate safety mechanisms directly into the persuader. For example, malicious claims can be included in the training data, with rewards designed to encourage the model to refuse to engage with such topics.
> - For high-stakes or sensitive topics, incorporate human review or escalation mechanisms to ensure outputs meet ethical and safety standards before delivery. Giving LLM agents full autonomy may lead to undesirable results.
>
> ---
>
> **Q4: Could the same results have been reached by exploiting high temperatures for LLM response generation and building a tree of discussions out of the generated options?**
>
> We agree that decision trees are commonly used in adversarial settings, and we are aware of related work in debate (**[1]**). It is a natural extension to combine decision tree–based strategies with ToM and RL for persuasion. However, we don’t think such methods will reduce compute cost. Instead, such approaches are more computationally expensive, as the persuader must generate multiple trajectories at each step. On the other hand, our attitude predictor is lightweight and provides decent accuracy (refer to **Sec. 4.3.2**) in assessing different perspectives.
>
> ---
>
> We value your insightful comments and suggestions, which have significantly contributed to the improvement of our paper. We hope that our detailed responses and additional experiments have effectively addressed your concerns. If you find that your concerns have been resolved, we would be very grateful if you could reconsider the review score.
>
> **References**
>
> **[1]** Strategic Planning and Rationalizing on Trees Make LLMs Better Debaters, ICLR26

---

> > ### Author Rebuttal · Reviewer_5VYx · 2026-04-03
> >
> > LLM vs human persuasion: I do agree with the authors that there are valid use-case for LLM persuasion as well. This could be stated more clearly in the paper. When LLM persuasion is the main objective, a study about alignment between humans and LLM in perceived persuasiveness is not very relevant, as this is but a link in the chain of persuasion and the study should have focused on LLM-human alignment over the actual persuasion. If human persuasion is the end goal, then the alignment study is relevant, but not sufficient. The distinction between these is crucial and a choice should be clearly articulated, with an approprioate set of experiments for that choice.
> >
> > Human attributes assigned to LLMs: sufficiently addressed if changes to the paper are made.
> >
> > Safeguards: You give a few options for safeguarding. These seem like sensible options. Nevertheless, releasing code of your experiments here, as you say you will, will release a persuader without any safeguards, leaving the safety and ethical issues to the end-user. To me, this is not enough effort from your side, on a crucial aspect of your technology.
> >
> > Other options: Perhaps I did not explain myself well enough, because my suggestion was to use trees instead of the persuader, rather than a tree of persuaders. The only thing you then need is a cost function for the tree splits that includes your light-weight attitude predictor and explore the tree in the direction most preferred attitude-wise. This is guaranteed to work well, as the ToM is missing and consequently there won't be forward thinking about reasoning, but I don't see why it could not also result in persuasive arguments if the attitude predictor is used effectively.

---

> > > ### Author Response · Authors · 2026-04-03
> > >
> > > We highly appreciate the reviewer's comments. We will refine the paper's presentation on "LLM vs human persuasion" and "Human attributes assigned to LLMs" as suggested. We will consider building upon our work in the near future, and the decision tree idea (we now understand the point) is very interesting and worth consideration.
> > >
> > > **Regarding the safety concern, we have some additional explanations:**
> > >
> > > We would like to clarify that the potential harmfulness of a persuader depends not only on the training method itself, but more critically on the data used during training. In our work, the dataset has been curated to exclude highly controversial, sensitive, or dangerous content, substantially limiting the risk of harmful behavior arising from our model.
> > >
> > > More fundamentally, we believe that mitigating risks at the data level is essential. Preventing the inclusion of malicious or harmful data sources is a more effective and principled approach to ensuring safety than placing the primary responsibility solely on algorithm designers. While our approach can be reused in different contexts, it is the combination of code and data that determines the system’s real-world impact.
> > >
> > > That said, we acknowledge the reviewer’s concern and will further emphasize in the revised manuscript the importance of responsible data curation, as well as provide clearer guidance on appropriate use.

---

### Official Review · Reviewer_QTb4 · 2026-03-16

**Soundness:** 3
**Presentation:** 3
**Significance:** 3
**Originality:** 2
**Overall Recommendation:** 4
**Confidence:** 3

**Summary:**

This paper proposes ToMAP, a framework that utilize RL to enhances LLM persuaders by incorporating 2 ToM modules: a counterclaim predictor (prompt-based, generates potential objections) and an opponent attitude predictor (BGE-M3 + MLP, estimates the persuadee's agreement level on counterclaims). With only 3B parameters, ToMAP outperforms much larger models including GPT-4o and DeepSeek-R1, with a 39.4% relative gain across multiple persuadee models and diverse corpora.

**Compliance With Llm Reviewing Policy:**

Affirmed.

**Final Justification:**

Concerns are fully resolved. I have raised my score.

**Key Questions For Authors:**

1. Have authors considered multi-step planning approaches (e.g., MCTS with binary persuasion success/failure as terminal reward)? The current 1-step lookahead may miss long-range strategic opportunities.
2. What happens when the same model instance serves as both persuader and persuadee? Does the persuader's advantage diminish when the opponent is (1) the same model as persuader (2) also ToM-aware?

**Limitations:**

1. Evaluation is limited to LLM-to-LLM persuasion. No human-in-the-loop experiments are conducted to validate real-world persuasiveness.
2. No built-in safety guardrails; the paper relies on external usage guidelines only.

**Strengths And Weaknesses:**

**Strengths**
1. Writing is clear, figures are well-designed, and the appendix provides helpful qualitative examples.
2. Comprehensive analysis covers repetition (Figure 3b), long-conversation trends (Figure 4), categorical persuasion strategy distribution (Figure 5), etc.
3. Apart from automated evaluation, human agreement metric is included to validate the attitude predictor (Section 5.1).
4. A 3B model outperforming GPT-4o and DeepSeek-R1 demonstrates strong practical value.

**Weaknesses**
1. The motivation is not fully convincing
	1. If the counterclaim predictor is not finetuned, the framework doesn't address the lack of diversity issue in base LLM. It only improves the counterclaim selection.
	1. Table 3 shows the identical performance between Long CoT and ToMAP (w/o att), is it a typo or counterclaim predictor is actually unnecessary?
1. The attitude predictor's limited accuracy (59.57% EM) introduces prediction bias that may be amplified through RL training, potentially leading to systematically suboptimal claim selection.
1. The paper lacks built-in safety mechanisms to prevent harmful or manipulative outputs.

**Note**: I am glad to raise my score depending on how authors ***persuade*** me regarding my doubts (especially weakness #1).

---

> ### Author Rebuttal · Authors · 2026-03-26
>
> We're very grateful for your constructive comments, and are glad for your acknowledgement of our paper’s strong results and thorough experiments. Below are responses to your suggestions and concerns.
>
> **Q1.1: How does counterclaim predictor address the lack of diversity issue in base LLM, if it’s not finetuned?**
>
> By diversity, we refer to the persuader’s ability to explore different directions and sub-arguments for a given issue. LLMs inherently possess this capability—they can readily “brainstorm” supporting evidence for a claim with reasonable quality. However, when engaged in multi-turn conversations, LLMs tend to focus on a single line of reasoning and repeatedly reinforce the initial argument (as illustrated in **Fig. 3.b** for the non-ToM persuader).
>
> To address this limitation, we introduce a counterclaim predictor that explicitly “reminds” the persuader of alternative perspectives. In this sense, our approach elicits the LLM’s existing knowledge rather than injecting new information. As a result, the diversity of generated arguments is improved without requiring fine-tuning.
>
> ---
>
> **Q1.2: What does the identical performance between Long CoT and ToMAP (w/o att) indicate?**
>
> We thank the reviewer for pointing this out. This is a typo, and we will correct that in the revised version. The actual result is listed below:
>
> | Method | CMV | Amthropic | args.me |
> | --- | --- | --- | --- |
> | Long CoT | 14.25 | 6.25 | 7.66 |
> | ToMAP (w/o att) | 15.57 | 9.55 | 10.45 |
> | ToMAP | 23.01 | 15.82 | 18.75 |
>
> While long CoT brings marginal benefit compared to the baseline, it underperforms both ToMAP (w/o att) and ToMAP. This indicates that both the counterclaim predictor and attitude predictor contribute to generating more effective rebuttals.
>
> ---
>
> **Q2: The attitude predictor's limited accuracy introduces prediction bias that may be amplified through RL training?**
>
> While it is true that the attitude predictor’s performance is not perfect, we believe the current setting is sufficient for the persuasion task for the following reasons:
>
> - Since attitude lies on a *continuous* spectrum, small deviations are often acceptable (e.g., when the true attitude is “agree” but the predictor outputs “partly agree”). Additionally, we find that the errors are largely random, rather than biased toward a particular direction.
> - As shown in **Table 3**, ToMAP-gt does not yield significant improvements over ToMAP, suggesting that further gains from improving attitude prediction accuracy are marginal.
> - Inferring the persuadee LLM’s attitude from context is inherently uncertain, making highly accurate prediction challenging, if not infeasible.
>
> ---
>
> **Q3: The paper lacks built-in safety mechanisms to prevent harmful or manipulative outputs.**
>
> Please refer to **Q3** of reviewer **5VYx for details about this question.**
>
> ---
>
> **Q4: Have authors considered multi-step planning approaches (e.g., MCTS with binary persuasion success/failure as terminal reward)?**
>
> Multi-step planning is commonly used in adversarial settings, and we are aware of related work in debate (**[1]**). It is a natural extension to combine decision tree–based strategies with ToM and RL for persuasion. However, such approaches are computationally expensive, as the persuader must generate multiple trajectories at each step.
>
> In our preliminary exploration, we designed a global reward to evaluate the outcome of an entire conversation. However, we found that incorporating the global reward had little impact on model performance, and thus we removed it from the final paper. One possible explanation is that persuasion is a task where local optima often align with the global optimum, making a one-step reward sufficient.
>
> ---
>
> **Q5:  What happens when the same model instance serves as both persuader and persuadee? Does the persuader's advantage diminish when the opponent is (1) the same model as persuader (2) also ToM-aware?**
>
> (1) This setting is unlikely to happen in our experiment, because trained persuaders are specialized LLMs, so we won’t use them as persuadees.
>
> (2) That’s a very interesting question. One can possibly train better persuadees with ToM, using techniques similar to GAN to make both agents evolve simultaneously. However, as discussed in **Sec 4.3**, simply providing ToM information without corresponding training is unlikely to yield meaningful improvements.
>
> ---
>
> **Q6: Evaluation is limited to LLM-to-LLM persuasion.**
>
> Please refer to **Q1** of reviewer **5VYx for details about this question.**
>
> ---
>
> We value your insightful comments and suggestions, which have significantly contributed to the improvement of our paper. We hope that our detailed responses and additional experiments have effectively addressed your concerns. If you find that your concerns have been resolved, we would be very grateful if you could reconsider the review score.
>
> **References**
>
> **[1]** Strategic Planning and Rationalizing on Trees Make LLMs Better Debaters, ICLR26

---

> > ### Author Rebuttal · Reviewer_QTb4 · 2026-04-04
> >
> > The necessity of counterclaim predictor is still unclear, as it's basically prompting technique. Besides, Long CoT baseline details are missing from the paper.
> > 1. How does it perform when you guide the Long CoT baseline to (a) propose counterclaims and (b) corresponding attitudes in the prompt?
> > 2. Do you use reasoning-finetuned versions of base model? I understand there is no "thinking" version of Qwen2.5, but can you try Qwen3 or 3.5 with thinking? If I understand this correctly, to report Long CoT and ToMAP w/o att results with another base model, you don't need to train the model.
> > 3. How do larger models (DS-R1, GPT-4o, etc.) w/ Long CoT perform? I wonder to what extent these models can solve the problem.

---

> > > ### Author Response · Authors · 2026-04-05
> > >
> > > The reviewer is correct that the counterclaim predictor is implemented via prompting; however, its role goes beyond prompt engineering. Unlike a single-pass prompt, it introduces an explicit intermediate reasoning step that is integrated into the agentic pipeline.
> > >
> > > **The counterclaim predictor is structurally necessary for enabling the subsequent attitude prediction module.** Specifically, attitude prediction in ToMAP is conditioned on explicitly enumerated counterclaims, rather than implicit or latent representations. Without this step, the model lacks a concrete set of opposing viewpoints to reason over, making attitude prediction unfeasible in practice. As shown in Eq. 9, the counterclaims $¬q$ are required for the attitude predictor.
> > >
> > > Regarding the experiments:
> > >
> > > (2) (3) We’d like to highlight that **Table 2** actually encompasses the model types mentioned by the reviewer, where Deepseek-R1 is a reasoning model, and GPT-4o and LLaMa-70B are large-scale models. We conclude that larger models typically utilize ToM information better in a zero-shot manner, but their gains with ToM are still limited, as they’re not trained specifically for persuasion.
> > >
> > > (1) In the main paper’s experiments, we give a general prompt to lead the model to “consider carefully how to formulate more convincing arguments”. We thank the reviewer’s suggestion to give elicit instrction in analogy with the two ToM modules to enable more fair comparison. The results are listed in the following table:
> > >
> > > | Persuader | Qwen2.5-3B | Gemma-3-27B | LLaMa3.1-70B | Deepseek-R1 | GPT-4o | Qwen2.5-3B+ToMAP |
> > > | --- | --- | --- | --- | --- | --- | --- |
> > > | w/o ToM info | 8.44 | 11.92 | 8.31 | 11.40 | 10.17 | 14.47 |
> > > | w/ Long CoT  | 8.73 | 12.58 | 9.19 | 11.46 | 12.10 | N/A |
> > > | w/ CoT for counterclaim | 9.55 | 12.90 | 8.95 | 12.20 | 11.97 | N/A |
> > > | w/ CoT for counterclaim & attitude | 9.17 | 14.37 | 9.49 | 14.18 | 13.86 | N/A |
> > > | w/ ToM info | 8.18 | 13.26 | 9.56 | 11.32 | 11.69 | 19.19 |
> > >
> > > From the tables, we can observe that prompting for counterclaim yields very limited, or even negative gains, while prompting for both do increases the performance. These results show that while using more specific prompts is helpful, it still underperforms the separate pipeline introduced in ToMAP.
> > >
> > > We will incorporate the additional experiment into the revised paper, and we hope these answers address your concerns.
> > >
> > > Authors

---

### Decision · Program_Chairs · 2026-04-30

**Decision:**

Accept (regular)

**Comment:**

The submission "ToMAP: Training Opponent-Aware LLM Persuaders with Theory of Mind" trains LLMs to be maximally persuasive to other LLMs using a combination of initial prompting and reinforcement learning techniques, motivated by theory of mind research. The results of the submission show that this does result in small models (3b parameters) persuasiveness increasing up to 40%.

Reviewers considered this work overall to be very interesting, technically sound, and the results noteworthy. Much of the discussion during the rebuttal phase has been concerned with the important meta-question of the author's plan for safe usage of this technology, discussing the proposed and already implemented safeguards, data filtering to concentrate the approach on persuasion respecting human debate preferences, concepts for multilayered deployment and the effect on harmful question elicitation. I do consider these questions to be important, but I also think that, with the existing text and the additions during the rebuttal phase, the authors provide a sufficient overview.

There are further a few comments regarding the clarity of writing that I hope the authors can address. Overall this work should be discussed at ICML 2026.